# Temporal Variation in Soil Resistance to Rill Erosion in Cropland of the Dry—Hot Valley Region, Southwest China

**Yi Wang** [1,2,3], **Xiaosong Qin** [2], **Yaping Kong** [4,*], **Dongdong Hou** [2] and **Ping Ren** [2]

1. Key Laboratory of Ministry of Education on Land Resources Evaluation and Monitoring in Southwest China, Sichuan Normal University, Chengdu 610066, China; wy@sicnu.edu.cn
2. College of Geography and Resources Science, Sichuan Normal University, Chengdu 610101, China; xiaosongqin@stu.sicnu.edu.cn (X.Q.); 20231101076@stu.sicnu.edu.cn (D.H.); renping@sicnu.edu.cn (P.R.)
3. Engineering Research Center for the Development of Farmland Ecosystem Service Functions, Chengdu 610068, China
4. China Academy of Transportation Sciences, Beijing 100029, China
* Correspondence: kongyp@motcats.ac.cn

**Abstract:** In croplands, soil erosion resistance varies with both natural processes and human disturbances. To clarify the temporal variation in soil erosion resistance, nine cropland plots with three treatments (continuous fallow, fallow after tillage and tillage with corn) were established in the dry–hot valley region of China. A total of 144 field runoff simulation experiments were conducted from May to October to measure the soil detachment rate ($D_c$), rill erodibility ($K_r$) and critical shear stress ($\tau_c$). The results revealed that the natural dry—wet alternation had little influence on the continuous-fallowed soil erosion resistance. On the other hand, the tillage disturbance that occurred in May sharply increased the $D_c$ and $K_r$ to 2.24 and 3 times that of the continuous-fallow treatment, respectively. Then, the erosion resistance could be enhanced with surface consolidation for the fallow-after-tillage treatment. However, after three months of fallow, the $K_r$ was still 89.5% of the fresh tilled soil. In contrast, crop growth could significantly improve aggregate stability and reduce the $K_r$ to 38.2% in August and even further to 23.7% in October compared to the fresh tilled soil. It could be concluded that crop growth is more efficient in enhancing erosion resistance than the mechanical effect. The above results would benefit from the accurate modeling of cropland soil erosion dynamics and guide agricultural management in dry–hot climate regions.

**Keywords:** soil erosion resistance; rill erodibility; field experiment; cropland; dry–hot climate region

## 1. Introduction

Soil erosion in croplands leads to the removal of the fertile topsoil layer, posing a significant threat to the sustainability of land resources and global food production [1]. To effectively control cropland soil losses, quantitatively describing the erosion processes and evaluating the associated risks is imperative. Soil erosion caused by water can generally be classified into three subprocesses, i.e., the detachment of soil particles from the soil masses by the force of raindrop splashing or flow scouring, the transportation of sediment by concentrated flows and the deposition processes [2]. During the concentrated flow rill erosion process, soil detachment is the initial step and occurs when the flow shear stress exceeds the threshold provided by soil erosion resistance [3]. According to the excess shear stress theory for rill erosion development, soil erosion resistance can be expressed using two parameters, i.e., the critical flow shear stress ($\tau_{cr}$) at which significant soil detachment begins and the rill erodibility ($K_r$) factor describing the increasing rate of soil detachment with shear stress once the critical value is exceeded [4,5]. Both parameters are important inputs for physical erosion models such as the Water Erosion Prediction Project (WEPP) [6,7]. Therefore, the measurement and quantification of soil erosion resistance are crucial for cropland erosion evaluation.

The resistance of soils to erosion is strongly affected by the physicochemical properties of surface soils [3]. For example, soil texture and rock fragment content play important roles in erosion resistance by influencing soil cohesion [8], and the soil detachment rate reportedly decreases with increasing clay content and increases with increasing silt content [9]. Analogously, soil bulk density and aggregate stability were reported to be negatively correlated with soil detachment capacity and rill erodibility [10,11]. Soil organic matter can influence topsoil structural stability and protect soil particles from being eroded; therefore, a higher soil organic matter content generally results in lower rill erodibility [5,12]. Moreover, the existence of some mineral elements also has a significant effect on the resistance of soils to erosive forces. For example, soils with higher iron and aluminum contents are generally more weathered and less erodible [13]. Based on the relationship between erosion resistance and soil properties, predictive equations have been built to calculate rill erodibility and critical shear stress [7].

In addition to the abovementioned soil indices, external environmental variables also impact intrinsic soil characteristics and lead to variability in soil erodibility [8]. For example, seasonal wetting and drying cycles lead to the natural reconsolidation of surface soil, which increases the bulk density and decreases the porosity, in turn reducing the rill erodibility [14]. Some scientists have reported that seasonal droughts reduce the stability of soil aggregates and thus result in a high detachment rate [15]. In addition, the vegetation root system, litter and its decomposed residues can enhance soil resistance by binding soil particles [16,17], altering soil surface roughness and soil organic matter [18]. Accordingly, temporal changes in vegetation coverage, root growth and crop decomposition during the growing season lead the seasonal variations in rill erodibility and critical shear stress [19]. Moreover, the effects of tillage disturbance on soil erosion potential should also be considered. Conventional tillage can sharply reduce the resistance of soils to erosion by breaking particle bonds, therefore resulting in high rill erodibility and lower critical shear stress [20]. Nevertheless, the tillage-disturbed erosion resistance would increase in the long term due to the consolidation and wetting–drying processes that increase the soil stability and cohesion strength [21,22]. Importantly, cropland soils are impacted by the combined effects of the aforementioned climate, vegetation and tillage disturbance, therefore resulting in complex variability patterns in soil erosion resistance that are not yet fully understood [23]. Soil erosion resistance factors are often difficult to quantify, especially in cases of intensive external disturbances or distinct temporal variations.

The dry–hot valley region, which mainly comprises the valleys along rivers in the Hengduan Mountains of Southwest China, is known for large-scale slope farming (Ministry of Water Resources (MWR), 2017) [24]. Intensive agricultural activities in slope croplands have led to serious soil erosion, which has been reported to account for nearly 60% of the total sediment generation in this area [25]. The dry–hot valley region is characterized by a seasonal cycle consisting of an extremely dry and hot season and a wet season with high-intensity rainfall [15], which in turn affects soil characteristics and erosion resistance [26]. Moreover, local tillage practices generally begin in the rainy season and result in a loose erodible soil layer, which contributes to a high erosion risk [27]. Consequently, the cropland soil erosion resistance is determined by a complicated interaction between soil properties and external factors that need to be understood. In recent years, local scientists have been mainly concerned with the high soil erosion risk related to tillage disturbance through runoff plot monitoring and field runoff simulation experiments [28,29]. Some studies simulated the dry—wet alternation process in the laboratory by saturating and drying soil and concluded that the dry—wet alternation effect reduced the erosion resistance [14]. Nevertheless, relatively little research has focused on the interaction effects of climatic rhythm, tillage disturbance and crop growth on cropland soil resistance. In particular, the responses of rill erodibility and critical shear stress are still rarely reported.

The existing knowledge gap about cropland soil resistance variation may lead to difficulties in modeling cropland soil erosion. Therefore, this study was conducted to reveal the variations in soil erosion resistance in typical croplands of the dry–hot valley region. We

hypothesized that the seasonal dry—wet alternation, tillage disturbances and crop growth would result in unique soil erodibility dynamics by interacting with each other and playing different roles. To evaluate this, field runoff simulation experiments were performed at different crop growth stages and the specific objectives were to (1) demonstrate the temporal dynamic characteristics of soil detachment, rill erodibility and critical shear stress under different field conditions; (2) evaluate the relationships between soil resistance factors and intrinsic soil characteristics; and (3) identify the main mechanisms that determine the dynamics of soil resistance. The results of this study clarify the interaction mechanism that determines cropland soil resistance dynamics. In turn, it could accurately calculate soil resistance indices and thus enhance the precision of soil erosion modeling in cropland. This study will be helpful in guiding erosion evaluation and conservation under dry–hot climate conditions in Southwest China and other similar regions worldwide.

## 2. Materials and Methods

### 2.1. Study Site

The experimental site is located in the Xiaotuanshan watershed in Dechang County, Sichuan Province, Southwest China (102°11′2″ E, 27°19′37″ N) (Figure 1a). This region has a typical subtropical monsoon climate with distinct dry and rainy seasons. The average temperature is 17.7 °C, with an annual sunshine duration of 2147 h and a frost-free period of more than 300 days [30]. The climate is warm temperate with the winter dry and a warm summer (Cwb according to the Koppen Geinger classification, https://koeppen-geiger.vu-wien.ac.at/ (accessed on 4 April 2024)). The total annual precipitation is recorded as 1150 mm, and 90% of the precipitation is concentrated within the wet season from May to October (Figure 2). Most torrential rains are of a short duration, which readily cause serious soil erosion risks. The dry season lasts from November of a given year to April of the following year, and very little rainfall and an extremely dry climate occur during this period [25]. The watershed is characterized by hilly terrain with elevations below 1500 m.a.s.l. and a relative elevation difference between 200 m and 300 m. The soils in the study site are derived from quaternary ancient red soils [31] and can be classified as Alisols according to the World Reference Base (WRB, 2015) [32] international soil classification system.

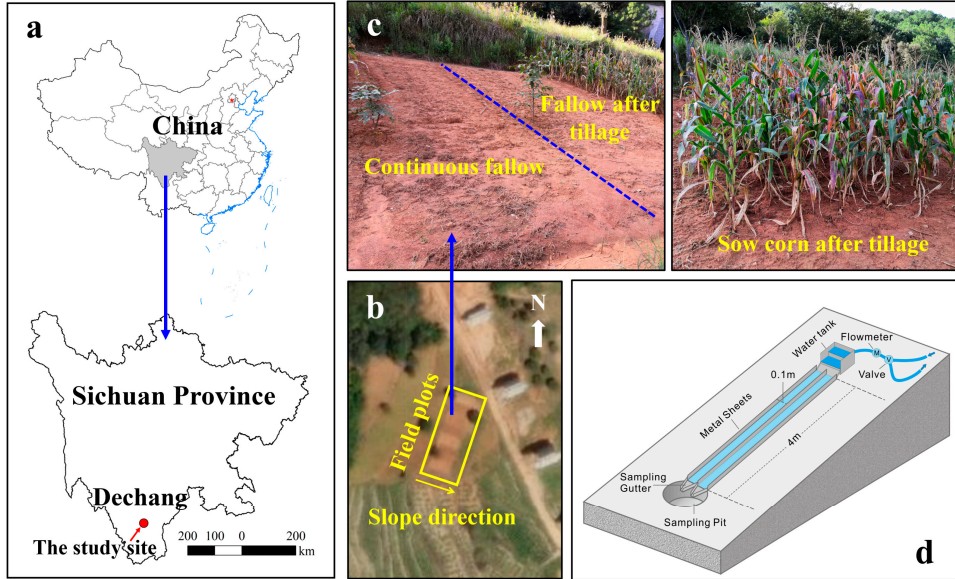

**Figure 1.** Study area and the experimental design. (**a**) the location of the study area, (**b**) field plots in Google Maps satellite image, (**c**) field plots of three different treatments, (**d**) the design of the runoff simulation experiment.

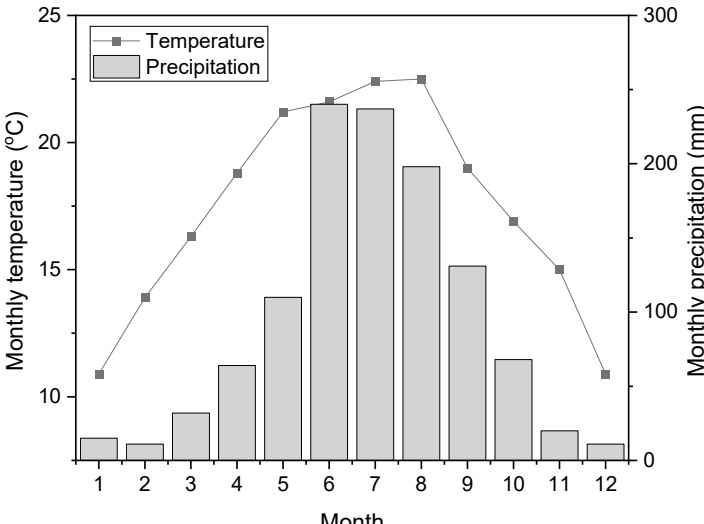

**Figure 2.** The average monthly temperature and precipitation in the study area.

The study site is located within the Anning River valley area, which is known as the main agricultural and grain-producing area in western Sichuan Province. Slope cropland is frequently adopted in the belt area between the valley and hills along both sides of the river. In some cases, sloping farmland can be found on gradients steeper than 25 degrees. Traditional hoeing tillage is the most commonly used tillage method [28]. Limited by the local climate and water conditions, these sloping farmlands are generally cultivated in the wet season and fallow in the dry season. Corn is a commonly planted local crop and is harvested once a year in most cases.

### 2.2. Design of Field Plots

The experimental site is a corn field in which corn has been cultivated for more than 10 years. The soil texture in the top 20 cm of the field plot is clay loam (USDA classification), with 34.33–36.73% clay, 21.60–24.67% silt and 41.00–42.00% sand. Nine experimental plots with a 45 m$^2$ area (9 × 5 m) were set up following one year of fallowing after the last corn harvest (Figure 1b). The plots were designed with different slope gradients and treatment combinations. Three slope gradient grades, i.e., 8 degrees (S1), 12 degrees (S2) and 15 degrees (S3), were selected according to the field terrain conditions of the local slope farmland [29]. Continuous fallow, fallow after tillage and sow corn after tillage were set to reflect the different conditions that influence soil characteristics and erosion resistance (Figure 1c and Table 1). Continuous fallow had been kept fallow since the corn was harvested in the previous year, and it was mainly used to quantify the effect of the dry—wet alternation on soil erosion resistance. The two tilled treatments were cultivated in early May before the beginning of the wet season and were used to represent the effect of the tillage disturbance and the following soil consolidation and crop growth on soil erosion resistance. Traditional downslope manual hoeing tillage practices were adopted, and the cultivation depth was approximately 20 cm. For the tillage-with-corn treatment, corn was sown in downslope rows in May at the beginning of the wet season. The row spacing was 50 cm, and the plant interval within each row was approximately 30 cm. The plant population in each plot was approximately 67,000 plants/hm$^2$. The plots were fertilized at the same time as the corn sowing with 600 kg/hm$^2$ of Stanley compound fertilizer (N:P$_2$O$_5$:K$_2$O = 15:15:15) and 5000 kg/hm$^2$ of decomposed manure, in accordance with the local requirements for corn growth [33]. To avoid disturbance to the soil surface, herbicide was used to control the weeds. In the fallow treatments, all the weeds were removed to eliminate the influence of plants on the soil detachment process and ensure that soil resistance is mainly affected by natural dry—wet alternation.

**Table 1.** Descriptions of the experimental plot treatments.

| Treatment | Description | Purpose |
|---|---|---|
| Continuous fallow | Kept fallowed after the last year's corn harvest | To reflect the influence of the natural dry–wet cycle on soil detachment and resistance. |
| Fallow after tillage | Tilled in May and then fallowed | To reflect the effect of tillage disturbances and the following consolidation effect on the soil resistance dynamics. |
| Tillage with corn | Tilled in May and corn was sown | To reflect the effect of tillage disturbances and crop growth on soil resistance dynamics. |

*2.3. Measurement of Soil Detachment*

Field runoff simulation experiments were conducted in 2019 to measure the soil detachment rates. As soil erosion mainly occurs in the wet season, three time phases covering the wet season were selected according to the growth stages of corn. The first series of experiments was performed at the beginning of the wet season in May (stage 01, beginning of seed imbibition), shortly following the tillage disturbance. The second series of experiments was conducted in the middle wet season of August at the corn growth stage 51 (beginning of tassel emergence). The third series of experiments was performed in October when the corn was harvested (stage 87, physiological maturity) at the end of the wet season.

The runoff simulation experiments were set in each of the abovementioned 9 field plots (Figure 1d). To ensure that the flowing water covered all the soil surfaces, a belt 4.0 m long and 0.1 m wide was used to simulate a rill according to the design of Cao et al. (2011) [34]. The rill length is also the same as in our previous hydraulic flume experiment about cropland soil detachment [20]. Iron sheets were used as rill boundaries and inserted 10 cm into the soil and 10 cm above the soil surface. Clean water pumped from a nearby pond was supplied at the rill head with unit width flow rates of 0.001 $m^2 s^{-1}$, 0.002 $m^2 s^{-1}$ and 0.003 $m^2 s^{-1}$. These flow rates were selected according to Cao et al. (2015) [35] and were within the range of the runoff generation capability recorded by local erosive storm studies [36]. The flow rate was controlled by a valve and a flowmeter and would be calibrated by collecting water flowing to a graduated container in a given period before the experiment. A pit was dug at the bottom of the rill to collect the runoff and sediments using sampling bottles of a specific volume. During the experiments, samples were taken every 1 min. To standardize the effect of testing the soil conditions during the erosion process, we referred to Cao et al. (2011) [34] and stopped the experiment if the scour depth was greater than 5 cm. If the depth was maintained as shallower than 5 cm, the test was stopped after 10 min. Nine slope gradient and flow rate combinations were applied for each treatment. Two parallel tests were conducted as replicates for each combination. To ensure similar initial soil surface conditions, each simulated rill was used only once, and all the experiments were performed at new sites. A total of 18 experiments were conducted on each treatment at one time. Considering that fresh tilled plots (fallow after tillage and sow corn after tillage) have essentially the same conditions, 36 experiments were conducted in May. In August and October, 54 experiments were performed each time. In total, 144 field runoff simulation experiments were conducted, and 1096 sediment samples were collected in this study.

The sediments collected during each test were oven-dried at 105 °C for 24 h to determine the sediment concentration and total sediment yield. Then, the soil detachment rate was calculated as the total sediment yield divided by the experiment time and soil surface area [37]:

$$D_r = M/(T \times A) \tag{1}$$

where $D_r$ is the soil detachment rate (kg $m^{-2} s^{-1}$), $M$ is the total sediment mass for each test (kg), $T$ is the experimental duration and $A$ is the rill bed area covered by flowing water ($m^2$).

### 2.4. Calculation of Rill Erodibility and Critical Shear Stress

In this study, rill detachment was described using excess shear stress models; that is, the net soil detachment in rills was calculated for cases in which the hydraulic shear stress exceeded the critical shear stress for the soil and when the sediment load was less than the sediment transport capacity [38]. Based on the flow rate and measured flow depth, the flow shear stress was calculated as follows:

$$\tau = \rho g h S \tag{2}$$

where $\tau$ (Pa) is the shear stress, $\rho$ (kg m$^{-3}$) is the water mass density, $g$ (m s$^{-2}$) is the gravity constant, $h$ (m) is the depth of the flow within the rill and $S$ is the tangent value of the slope gradients.

The relationship between the soil detachment rate and sediment load can be described by the following equation [38]:

$$D_r = D_c(1 - \frac{G}{T_c}) \tag{3}$$

where $D_r$ is the detachment rate (kg m$^{-2}$ s$^{-1}$), $D_c$ is the detachment capacity by the rill flow (kg m$^{-2}$ s$^{-1}$), $G$ is the sediment load (kg m$^{-2}$ s$^{-1}$) and $T_c$ is the transport capacity of the flow (kg m$^{-2}$ s$^{-1}$). $D_c$ can be expressed as follows:

$$D_c = K_r (\tau - \tau_{cr}) \tag{4}$$

where $D_c$ is the detachment capacity (kg m$^{-2}$ s$^{-1}$), $K_r$ is the rill soil erodibility (s m$^{-1}$), $\tau$ is the flow shear stress that acts on soil particles (Pa) and $\tau_{cr}$ is the critical shear stress (Pa). As clean water without sediment was used in the experiments, the G term in Equation (3) could be set to zero, and $D_r$ was thus equal to $D_c$ [4]. Thus, $K_r$ and $\tau_{cr}$ could be calculated through a linear regression equation: $D_r = b_\tau + a$, where $K_r = b$ and $\tau_{cr} = -a/b$ [39].

### 2.5. Measurement of Soil Properties

In addition to the detachment rate and resistance indexes, the soil properties were also measured while the field runoff simulation experiments were conducted. For each treatment, 9 soil samples were taken randomly from the topsoil close to the testing rills to measure the physical and chemical properties of the soils. The soil bulk density was determined by the oven-drying method. The mean weight diameter of water-stable aggregates (MWD, mm) was measured by the wet-sieving method [40]. The organic matter content was determined by the Walkley and Black method. Moreover, the soil shear strength was measured under actual soil moisture conditions (8%, 12% and 18% in May, August and October, respectively) using a 14.10 vane pocket tester equipped with a three-head CL-100 (Eijkelkamp Soil & Water, Nijverheidsstraat, The Netherlands).

### 2.6. Statistical Analysis

The soil properties, detachment rate and resistance indexes were compared among different tests using analysis of variance (ANOVA) and least significant difference (LSD) multiple range tests. Correlation analyses were performed to describe the relationships between soil resistance and soil properties. Regression analyses were adapted to quantify the relationships between soil erodibility and influencing factors and to build a prediction model. The efficiency of the regression results was evaluated using the coefficient of determination. The above results were reported at the $p < 0.05$ significance level. All the analyses and graphical displays were made using the SPSS statistical software package (version 20.0) and Origin 8.5 (OriginLab, Northampton, MA, USA).

## 3. Results

### 3.1. Surface Soil Properties for Different Treatments

Table 2 shows the average soil property indexes representing the three treatments at different stages measured when conducting the experiments. For the continuous-fallow treatment, both the soil bulk density and organic matter content showed little variation during the wet season. The MWD values derived for the continuous-fallow treatment decreased by 21.69% from May to October. On the other hand, the shear strength of the surface soil increased by 17.48% throughout the research period. As the soils were disturbed by the tillage activity, the results showed significantly lower soil bulk densities shortly after cultivation (Table 2). For the fallow-after-tillage treatment, the bulk density, MWD and soil shear strength metrics were still lower than those of the continuous-fallow treatment in August due to the tillage disturbance. However, in October, compared with those in May, all three soil property indexes of the fallow-after-tillage treatment increased by 19.71%, 21.83% and 34.70%, respectively, and reached the same levels as those of the continuous-fallow treatment. Under the tillage-with-corn treatment, the disturbed soil structure recovered faster than that measured under the fallow-after-tillage treatment (Table 2). The MWD values measured in October for the tillage-with-corn treatment were 1.39 and 1.52 times higher than those for the fallow-after-tillage and continuous-fallow treatments, respectively. The bulk density and shear strength, however, were always lower than those of the two fallowed treatments during the corn growth period. Furthermore, there is a positive correlation ($r = 0.758$, $p = 0.049$) between the above two indexes.

**Table 2.** Soil properties obtained for three treatments in different stages.

|  | Treatment | MWD [a] (mm) | Bulk Density (g cm$^{-3}$) | Organic Matter Content (g kg$^{-1}$) | Shear Strength (kPa) |
|---|---|---|---|---|---|
| May | Continuous fallow | 1.66 | 1.42 a | 19.65 | 35.12 |
|  | Fresh tilled | — | 1.11 b | 20.01 | — |
| August | Continuous fallow | 1.34 b | 1.38 a | 20.92 | 36.08 a |
|  | Fallow after tillage | 1.14 b | 1.33 b | 20.02 | 26.12 b |
|  | Tillage with corn | 1.81 a | 1.28 b | 23.24 | 25.16 b |
| October | Continuous fallow | 1.30 b | 1.40 | 20.05 | 42.56 b |
|  | Fallow after tillage | 1.42 b | 1.42 | 22.42 | 40.00 b |
|  | Tillage with corn | 1.98 a | 1.37 | 24.71 | 24.04 a |

[a] MWD and shear strength were not measured in May due to the destruction of the surface soils. Different letters (a,b) indicate significant differences within each column at the $p < 0.05$ level.

### 3.2. Variation in Soil Detachment Rate under Different Treatments

The 72 average soil detachment rates measured at nine slope gradient and flow rate combinations were summarized according to the different treatments and growth stages (Figure 3). Generally, the average soil detachment rate for the continuous-fallow treatment changed little with time. In May, the average detachment rate of the fresh tilled soil (0.051 kg m$^{-2}$ s$^{-1}$) was approximately three times higher than that of the continuous-fallow treatment (0.017 kg m$^{-2}$ s$^{-1}$). In August, the average soil detachment measured under the fallow-after-tillage and tillage-with-corn treatments sharply decreased by 56% and 74%, respectively, compared to that measured for the fresh tilled soil in May. The average soil detachment rates measured under different treatments were ranked as fallow after tillage (0.022 kg m$^{-2}$ s$^{-1}$) > continuous fallow (0.016 kg m$^{-2}$ s$^{-1}$) > tillage with corn (0.013 kg m$^{-2}$ s$^{-1}$). After having been kept under fallow conditions for approximately 5 months, when measured in October, the fallow-after-tillage treatment had a similar soil detachment rate (0.016 kg m$^{-2}$ s$^{-1}$) as the continuous-fallow treatment (0.015 kg m$^{-2}$ s$^{-1}$). Moreover, the growth of the corn further decreased the soil detachment rate and resulted in the lowest average soil detachment rate value (0.010 kg m$^{-2}$ s$^{-1}$) among all the tests in October.

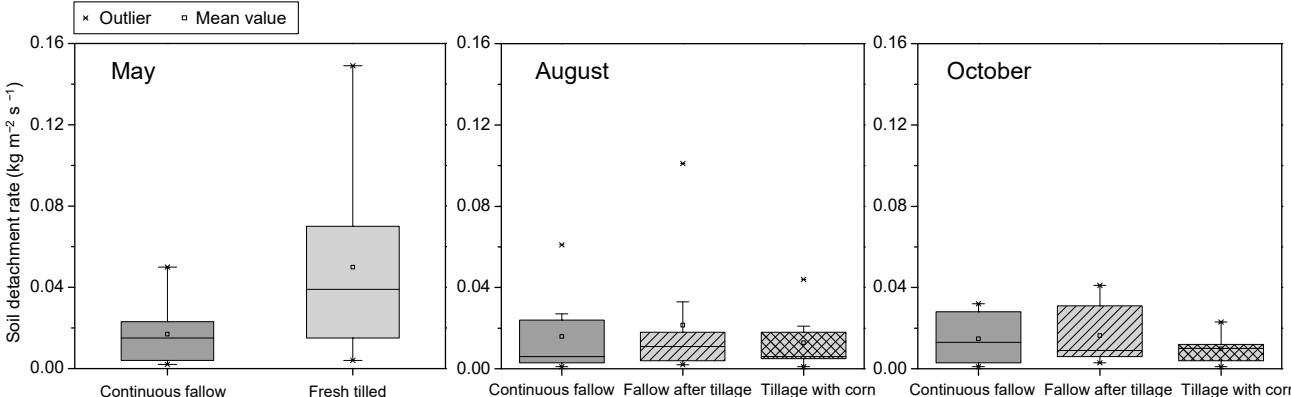

**Figure 3.** Temporal variation in soil detachment rates of the different treatments.

Further analyses were conducted by examining the soil detachment rates on different slope gradients (Figure 4). Figure 4 illustrates that the average soil detachment rates measured on the freshly cultivated plots were higher than those measured on the continuous-fallow treatment plots, but this difference was not significant on the low-gradient slopes (S1). As the slope became steeper (S2 and S3), the difference between the continuous-fallow and the newly tilled values sharply increased (ANOVA, $p = 0.035$). In the August and October tests, the average soil detachment rates measured for the three treatment plot types were similar for slope gradients lower than 12 degrees. The difference between the tillage-with-corn and the other two treatment plots was significant under the S3 slope gradient (ANOVA, $p = 0.029$).

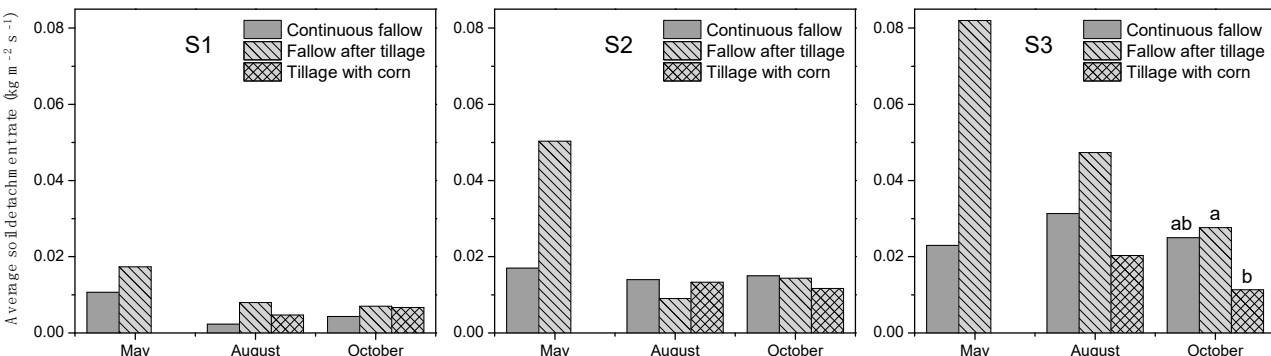

**Figure 4.** Average soil detachment rates summarized for different slope gradients. Different letters indicate significant differences within the three treatments at the $p < 0.05$ level.

### 3.3. Variations in the Erosion Resistance Indexes under Different Treatments

Based on the soil detachment rate and flow shear stress values measured at different flow rate and slope gradient combinations, the erosion resistance indexes were calculated according to Equation (4), and the equations are listed in Table 3. The measured rill erodibility (the equation coefficients, $K_r$) values showed a range of variation between 0.0018 s/m and 0.0076 s/m, and the critical shear stress (the intercepts, $\tau_c$) values ranged from 3.94 Pa to 6.68 Pa. According to Table 3, the $K_r$ values for the continuous-fallow treatment showed relatively small variation from May to October despite the slightly higher values in August (Table 3). Consistent with the soil detachment rate trend, the $K_r$ obtained for the fresh tilled soil was the highest in May. Approximately 3 months later, the $K_r$ for the fallow after tillage was reduced by 10.53% compared with the newly tilled value. As the rainy season continued, the $K_r$ of the fallow-after-tillage treatment was further reduced by 39.71% in October. Compared with the other two treatments, the rill erodibility of the tillage with corn was the lowest during the corn-growing season. In August, the $K_r$ for the sown corn soil sharply decreased to 38.2% of that of the fresh tilled soil, and the

ratio further decreased to 23.7% in October. For the critical shear stress ($\tau_c$) values, the continuous-fallow treatment showed a higher $\tau_c$ than the other two treatments, with the trend slightly increasing in August and decreasing in October. The $\tau_c$ value obtained for the freshly tilled soil in May was the highest during the observation period. Afterward, the $\tau_c$ values of the fallow after tillage and tillage with corn generally decreased with time from May to October, and the tillage with corn had a faster downward trend. Compared with May, the $\tau_c$ value of the tillage with corn decreased by 15.81% in August and 40.66% in October (Table 3).

**Table 3.** Calculation of resistance based on the shear stress equation.

| Growing Season | Treatment | Shear Stress Equation [a] | $R^2$ | n | SIG. |
|---|---|---|---|---|---|
| May | Continuous fallow | $D_r = 0.0034\,(\tau - 6.62)$ | 0.735 | 9 | 0.003 |
| | Fresh tilled | $D_r = 0.0076\,(\tau - 6.64)$ | 0.741 | 9 | 0.003 |
| August | Continuous fallow | $D_r = 0.0042\,(\tau - 6.86)$ | 0.883 | 9 | 0.000 |
| | Fallow after tillage | $D_r = 0.0068\,(\tau - 6.47)$ | 0.762 | 9 | 0.002 |
| | Tillage with corn | $D_r = 0.0029\,(\tau - 5.59)$ | 0.723 | 9 | 0.004 |
| October | Continuous fallow | $D_r = 0.0035\,(\tau - 5.51)$ | 0.866 | 9 | 0.000 |
| | Fallow after tillage | $D_r = 0.0041\,(\tau - 5.15)$ | 0.824 | 9 | 0.001 |
| | Tillage with corn | $D_r = 0.0018\,(\tau - 3.94)$ | 0.678 | 9 | 0.006 |

[a] $D_r$ and $\tau$ represent the detachment rate (kg m$^{-2}$ s$^{-1}$) and shear stress (Pa), respectively.

### 3.4. Factors Influencing Soil Resistance

The soil resistance was closely related to the soil properties. The Pearson correlation analysis indicated that the MWD and organic matter content were significantly negatively related to the soil detachment rate (Table 4). Notably, both the soil bulk density and the shear strength were not significantly related to the soil detachment rates. This was mainly due to the low bulk density and shear strength of the corn-planted soils, which also showed the lowest detachment rates among all the treatments. After removing the corn plot values, the correlation coefficients between the soil detachment rate and the bulk density were calculated as $-0.827$ and $-0.956$, respectively. In addition, rill erodibility was negatively related to the MWD, organic matter content and bulk density. Among the three indicators, the MWD was best correlated with rill erodibility. Similarly, when the values of the bulk density and shear strength for the corn-planted soils were deleted, the rill erodibility was also significantly negatively correlated with the two indicators of the soil properties, and the correlation coefficients were $-0.915$ and $-0.860$, respectively.

**Table 4.** The results of correlation analysis between the soil detachment rate or rill erodibility and soil properties.

| Parameter | | MWD [a] | Bulk Density | Organic Matter Content | Shear Strength |
|---|---|---|---|---|---|
| Soil detachment rate | r | $-0.827$ * | 0.048 | $-0.788$ * | 0.166 |
| | *p*-Value | 0.022 | 0.919 | 0.035 | 0.723 |
| | n | 7 | 7 | 7 | 7 |
| Rill erodibility | r | $-0.860$ * | $-0.104$ | $-0.629$ | 0.057 |
| | *p*-Value | 0.013 | 0.825 | 0.130 | 0.904 |
| | n | 7 | 7 | 7 | 7 |

[a] The fresh tilled values were not included due to the disturbed condition of the soils. The symbol * indicates significant levels at $p < 0.05$.

To further explore the mechanisms that govern soil erosion resistance, the detailed soil detachment rates measured during the experimental process were summarized for each treatment. Figure 5 shows that the soil detachment rates were generally high at the beginning of each experiment and then declined with the experimental duration. Power functions could be developed to describe the soil detachment rate trend during

the experimental process (Figure 5). The coefficient and fitness curve values of the freshly tilled soil were the highest in May. In August, the fallow-after-tillage treatment had the highest fitness curve; however, the coefficients of the continuous-fallow treatment were 1.66 times and 1.51 times higher than those of the fallow-after-tillage and tillage-with-corn treatments, respectively. On the other hand, the lowest function coefficient was measured for the tillage-with-corn treatment compared to the other treatments. Analogously, the soil detachment rates measured for the three treatments showed similar dynamic trends in October (Figure 4).

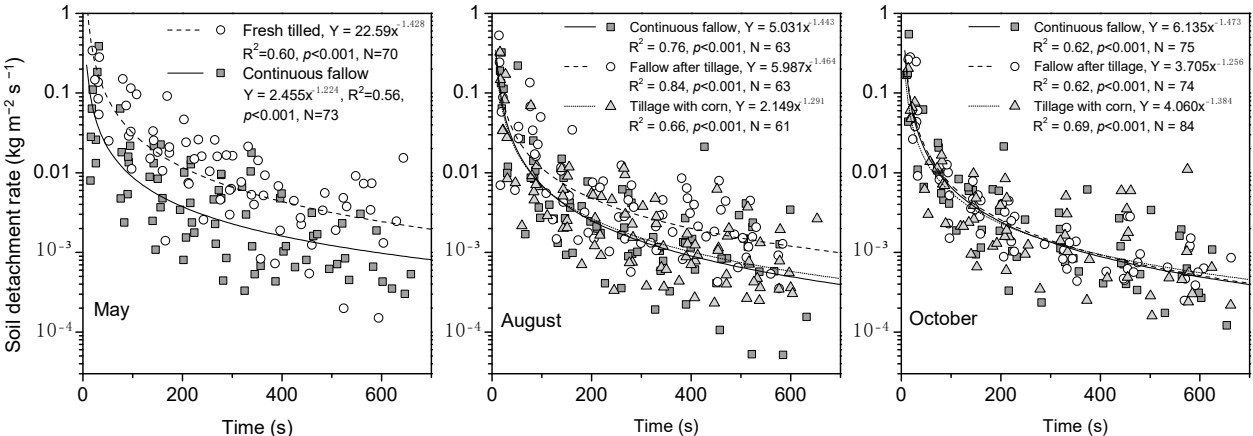

**Figure 5.** Soil detachment rate dynamics during the experimental process.

## 4. Discussion

### 4.1. The Effect of Natural Dry—Wet Alternation on Soil Erosion Resistance

For the continuous-fallow treatment, in which soil properties were mainly determined by the natural climate rhythm, the bulk density and the organic matter were almost unchanged with time. On the other hand, the decreasing MWD indicated the breakdown of surface soil aggregates as the dry season turned into the wet season. This might be explained by the differential swelling effect [40] during the wetting process of the local soil, for which the parent material is the quaternary ancient red soils with high expansibility [15]. Meanwhile, the raindrop impact effect would also account for the mechanical breakdown of soil aggregates [40,41]. In turn, the dispersion of soil particles contributes to the formation of structural crust with high tensile strength [42]. This process could explain the increasing shear strength trend shown for the continuous-fallow soils during the experimental period. The above results likely revealed an opposite dynamic trend between the aggregate stability and shear strength of the bare soil surfaces in this study. This result is different from previous studies under vegetation recovery conditions, in which the soil shear strength was mainly determined by the soil particle binding process and thus showed a similar trend with aggregate stability [19,43].

As has been reported by previous studies, higher soil aggregate stability and shear strength enhance erosion resistance [9,11]. Our dataset also indicated that the MWD and shear strength were negatively related to the soil detachment rate and rill erodibility (Table 4). Notably, the erosion resistance of the continuous-fallow soil seems to have been influenced by the abovementioned opposite trend between the MWD and shear strength. This might be the reason for the unchanged average soil detachment rate and rill erodibility from May to October in the continuous-fallow treatment. Considering that continuous-fallow soil is mainly affected by the natural dry—wet cycle, it could be concluded that natural dry—wet alternation alone would not lead to a dramatic change in soil erodibility in the study area.

### 4.2. Tillage Effect on Soil Erosion Resistance under Dry—Wet Alternating Conditions

Unlike natural dry—wet alternation, cultivation activity could dramatically change soil properties, therefore making the surface soil loose and leading to a decrease in bulk density and the destruction of soil aggregates [8]. This could be confirmed by the decreased bulk density of the fresh tilled soil compared with the continuous-fallow soil. Accordingly, the markedly higher average soil detachment rate and rill erodibility for the fresh tilled soil reflected that tillage disturbances broke the soil particle bonds, making the soil particles easily detached by the flowing water [20]. The highest coefficient and fitness curve values in Figure 4 imply that the high erodible potential of the tilled soil would exist throughout the detailed soil detachment process. Furthermore, the rising increment of soil detachment for the fresh tilled soil in Figure 3 shows that the increasing slope gradient enhances the flow energy, which in turn could detach more soil particles that have been disturbed by tillage [12].

When tilled soil is exposed to natural rainfall, the raindrop impact effect and soil particle weight lead to the consolidation of loose surface material, the reshaping of soil aggregates and the formation of soil crust [22,44]. This could explain the increasing trend in the MWD, bulk density, and shear strength from May to October of the tilled soil with fallow (Table 2). Along with the changing soil properties, the detachment rate and rill erodibility for the fallow-after-tillage treatment also decreased from May to October. This is consistent with previous studies showing that the erodibility of tilled soil is reduced due to consolidation and crust effects during the wet season [22,44]. Notably, the rill erodibility for the fallow after tillage in August is still 89.5% of that of fresh tilled soil and 1.62 times higher than that of continuous-fallow soil. This differs from the hydraulic flume experiments of Liu et al. (2024) [44], who reported that the rill erodibility for silt loam loess soil with 12.7% clay and 55.3% silt could decrease by more than 80% after approximately 100 days since the tillage disturbance. The slow decrease rate of rill erodibility due to the natural dry—wet alternation process might reflect the abovementioned swelling effect of the local soil, which leads to a relatively low MWD, bulk density and shear strength in August. Considering the high precipitation in this time stage, the weak erosion resistance indicated that the erosion risk caused by the tillage disturbance was still serious even after three months.

### 4.3. The Effect of Crop Growth on Soil Erosion Resistance

The significantly higher MWD for the tillage-with-corn soil (Table 2) might reflect the effect of the vegetation root system and residues, which can promote soil aggregation by stabilizing and binding soil particles [17,45]. Moreover, manure fertilizer might also explain the high MWD values [46]. The relatively high organic matter for the tillage-with-corn soil reflected this effect, through which the soil aggregate stability could be improved [47,48]. Unlike the results of the abovementioned fallow-after-tillage treatment, the tillage-with-corn soil bulk density and shear strength did not follow the same increasing trend as that of the MWD. The relatively lower bulk density and shear strength for the corn growth soil might mainly be due to crop coverage. The corn leaf development and stem elongation could intercept rainfall and reduce raindrop energy, thereby protecting the soil surface from being impacted by raindrops and weakening the consolidation and compaction effect [42].

As determined by the soil aggregate stability, both the detachment rates and rill erodibility of the tillage-with-corn treatment significantly decreased only three months after the tillage disturbance. Notably, the strong erosion resistance for the corn-planted soils is mainly correlated with the MWD and organic matter rather than the bulk density and shear strength. This implies that crop growth affects soil erosion resistance differently than the abovementioned natural wet—dry alternation induced a mechanical effect on the soil crust and consolidation process. Because the tillage-with-corn soil showed a lower detachment rate and rill erodibility than the fallow-after-till soil, it seems that crop growth and manure application would be more efficient in enhancing erosion resistance than the natural wet—dry alternation process. This is consistent with the findings of Cosentino et al. (2006) [48], who reported that the addition of organic matter had a greater impact on

aggregate stability than dry–wet cycles. Therefore, suitable agricultural management, such as rotation and fertilization measures, would be critical in easing erosion from cropland in the dry–hot valley region.

*4.4. The Response of Critical Shear Stress*

The measured $\tau_c$ values showed a range of variation between 3.94 Pa and 6.68 Pa (Table 3). These values are much higher than those measured by Su et al. (2019) [26] using laboratory flume experiments on similar soils ($\tau_c < 1$ Pa). This may reflect that concentrated flow energy is dissipated by the rough rill bed under field conditions; therefore, more shear stress is needed for rill initiation [5]. In fact, the $\tau_c$ values derived in this study were within the range of critical shear stress reported by Laflen et al. (1991) [6] for clay loam cropland soils based on field rainfall and runoff simulation experiments. Nevertheless, the dataset of this study did not support the idea of an increasing critical shear stress with decreasing rill erodibility as reported by laboratory hydraulic flume experiments [26,49]. Meanwhile, the high critical shear stress for the freshly tilled soil also differs from our previous flume experiment in which the disturbed soils showed lower $\tau_c$ values than undisturbed soils [20]. One reason for the above discrepancy might be the feedback effect between the eroding rill bed and flow shear stress. That is, an intensively eroded rill or the freshly tilled loosen materials would result in an undulating soil surface, thus reducing the flow velocity, which is negatively related to the flow depth in our experiments. According to Equation (2), a relatively deep flow depth results in a higher flow shear stress for rill erosion formation. Another reason could be attributed to environmental factors such as soil moisture, which were at their lowest in May and increased during the wet season. The negative relationship between critical shear stress and soil moisture reported by Li et al. (2022) [50] might account for the higher critical shear stress in May and lower values in October.

The above results indicated the complexity of the temporal dynamics of critical shear stress. In fact, many studies have shown that rill erodibility and critical shear stress may not necessarily exhibit a direct inverse relationship [51,52]. A review by Knapen et al. (2007) [3] found that tillage practices clearly affect rill erodibility but not critical shear stress, and the two indexes are not related to each other. Similar to our study, West et al. (1992) [53] concluded that the critical shear stress for consolidated soil was approximately 40% lower than that for fresh tilled soil. The complex temporal trend of critical shear stress in this study might confirm the opinion of Knapen et al. (2007) [3]; that is, rill erodibility seems to be a more appropriate parameter than critical shear stress to explain the variation in erosion resistance.

## 5. Conclusions

Cropland soil detachment rates were measured on three different treatments (continuous fallow, fallow after tillage and tillage with corn) in three temporal stages in the dry–hot climate region of Southwest China. Our dataset indicated that the natural dry—wet alternation led to an inverse change between aggregate stability and shear strength treatment, thus resulting in low temporal variations for continuous-fallow soil erosion resistance. In the case of tillage disturbance, both the soil detachment rate and rill erodibility sharply increased and then declined gradually with increasing bulk density and shear strength due to the mechanical effect of soil crust and consolidation. Nevertheless, we found that the rill erodibility is still 1.62 times greater than that of continuous-fallow soil even three months after tillage disturbance if no conservation measures are taken. In contrast, crop growth could effectively reduce the detachment rate and rill erodibility at the same time. This is mainly through the improvement in soil organic matter and aggregate stability rather than through the consolidation effect. It reflects the importance of plant protection in reducing rill erosion, and agricultural management, such as crop rotation or intercropping, should be applied to enhance cropland coverage. Compared with rill erodibility, our experimental data showed an inconsistent complex temporal trend of critical shear stress. In future

studies, further experiments should be conducted to reveal the dynamic mechanism of critical shear stress by considering the complexity of soil and environmental factors.

**Author Contributions:** Conceptualization, Y.W. and Y.K.; Data curation, X.Q.; Formal analysis, Y.K.; Funding acquisition, Y.W. and P.R.; Investigation, Y.K.; Methodology, Y.W.; Project administration, Y.W. and P.R.; Resources, Y.W.; Software, D.H.; Supervision, Y.K.; Validation, Y.W. and P.R.; Visualization, Y.K.; Writing—original draft, Y.W.; Writing—review and editing, Y.K. All authors have read and agreed to the published version of the manuscript.

**Funding:** This work was supported by the Sichuan Science and Technology Program (grant number 2023NSFSC1979) and the National Natural Science Foundation of China (grant number 41807077).

**Data Availability Statement:** The data that support the findings of this study are available from the corresponding author upon reasonable request.

**Acknowledgments:** The authors wish to express their appreciation to Yuan Xie and Jilun Yang for their assistance in the field experiment.

**Conflicts of Interest:** The authors declare no conflicts of interest.

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
