# Peer review of "Temporal Variation in Soil Resistance to Rill Erosion in Cropland of the Dry—Hot Valley Region, Southwest China"

_land, doi:10.3390/land13040546_

Round 1

Reviewer 1 Report

Comments and Suggestions for Authors

In croplands, soil erosion resistance varies with both natural processes and human disturbances. To clarify the temporal variation in soil erosion resistance, nine cropland plots with three treatments (continuous-fallow, fallow-after-tillage and tillage-with-corn) were established in the dry-hot valley region of China. A total of 144 field runoff simulation experiments were conducted from May to October to measure the soil detachment rate (Dc), rill erodibility (Kr) and critical shear stress (τc). The results of this study will be helpful in guiding erosion evaluation and conservation in dryhot climate conditions in Southwest China and other similar regions worldwide.

The logic of the article is clear, the data is detailed, but there are still some defects, the standardization of the format is not yet in place, it is suggested to modify, the specific amendments are as follows:

Abstract

1. P.1, line 18 - line 21 The research results presented in this part lack data support and are not intuitive enough, so it is suggested that necessary data should be appropriately supplemented in this part.

Introduction

1. The existing studies need to be further summarized. The logic of the whole introduction is not clear enough, and the language expression is not concise enough.

2. P.3, line 103 - line 113. The introduction of the existing research and shortcomings is not enough, and the summary of the importance and innovation of this research is not enough.

Materials and methods

1. P.5, line 170 - line 171 A belt 4.0 m long and 0.1 m wide was used to simulate a rill. What is the basis for the setting? 

2. P.5, line 178 - line 179 What is the setting basis for stopping experiment when the scour depth was greater than 5 cm or the time reaches 10 minutes? Is 10 minutes too long and will it affect the results of the soil detachment rate measurement?

Results

1. P.7 Generally, the position of the figure should be after the paragraph describing the figure. For example, the position of Figure 2 is wrong. Please check the full text carefully.

2. P.8 In Figure 3, it is suggested to label the analysis results of significance differences, which can be more intuitive.

3. P.8 Table 3 only lists the correlation coefficient of the linear regression equation, and there is no significance test result.

4. P.9, line 336.- line 337. The author did not describe the results of the significance test in Figure 4. How to show that the power function can be well used to simulate the changing trend of soil detachment rate.

Discussion

1. The discussion should focus on the purpose of the research, while parts 4.1-4.3 are a bit verbose, and should focus on each section to highlight the key points and further highlight the novelty of the research.

Comments on the Quality of English Language

None

Author Response

Issues raised by Reviewer 1:

Abstract

  1. P.1, line 18 - line 21 The research results presented in this part lack data support and are not intuitive enough, so it is suggested that necessary data should be appropriately supplemented in this part.

Reply and revision: Thanks for your comments and suggestions. We have made revision and added data in L20 “to 2.24 and 3 times that of the continuous-fallow treatment, respectively”.

Introduction

  1. The existing studies need to be further summarized. The logic of the whole introduction is not clear enough, and the language expression is not concise enough.

Reply and revision: Thanks for your comments and suggestions. We have improved the introduction section according to your and other reviewer’s comments. The sentences in L50-57, L63-64, L73-76, L83-85, L92 and L101-102 have been revised to make the language clearer and more concise.

  1. P.3, line 103 - line 113. The introduction of the existing research and shortcomings is not enough, and the summary of the importance and innovation of this research is not enough.

Reply and revision: Thanks for your comments. We have improved this section according to your and other reviewer’s comments. The existing research shortcomings were highlighted in L105-106. The importance and innovation have been more clearly expressed in L116-119.

Materials and methods

  1. P.5, line 170 - line 171 A belt 4.0 m long and 0.1 m wide was used to simulate a rill. What is the basis for the setting? 

Reply and revision: Thanks for your comments. we have added more contents to explain the design of the rill length and width in L190-192: according to the design of Cao et al. (2011) [34]. The rill length is also same to our previous hydraulic flume experiment about cropland soil detachment [20]. Two references have been added to support the setting.

  1. P.5, line 178 - line 179 What is the setting basis for stopping experiment when the scour depth was greater than 5 cm or the time reaches 10 minutes? Is 10 minutes too long and will it affect the results of the soil detachment rate measurement?

Reply and revision: Thanks for your comments. We have added more contents to explain this issue in L201-203: To standardize the effect of testing soil conditions during the erosion process, we referred to Cao et al. (2011) [34] and stopped the experiment if the scour depth was greater than 5 cm. The 10 min time duration is mainly adopted in the case of the weak erosion at low flow rates. As the scour depth is less than 5cm throughout the experiment, it would not result in an uneven surface condition. Therefore, the results of the 10 min experiments would be comparable to those of the shorter time durations.

Results

  1. P.7 Generally, the position of the figure should be after the paragraph describing the figure. For example, the position of Figure 2 is wrong. Please check the full text carefully.

Reply and revision: Thanks for your suggestion. We have checked all figures and made revisions accordingly.

  1. P.8 In Figure 3, it is suggested to label the analysis results of significance differences, which can be more intuitive.

Reply and revision: Thanks for your suggestion. We have revised the figure (Figure 4 in the revised manuscript) and have added significance differences.

  1. P.8 Table 3 only lists the correlation coefficient of the linear regression equation, and there is no significance test result.

Reply and revision: Thanks for your comments. We have revised Table 3 and added the significance test information.

  1. P.9, line 336.- line 337. The author did not describe the results of the significance test in Figure 4. How to show that the power function can be well used to simulate the changing trend of soil detachment rate.

Reply and revision: Thanks for your comments. We have revised the figure (Figure 5 in the revised manuscript) and added the significance level information for each power equation.

Discussion

  1. The discussion should focus on the purpose of the research, while parts 4.1-4.3 are a bit verbose, and should focus on each section to highlight the key points and further highlight the novelty of the research.

Reply and revision: Thanks for your comments and suggestions. We have improved the discussion section according to your and other reviewer’s comments. Contents in L388-389, L398-399, L406-407, L412-413, L421-426, L436-438, L454-456, L463-465 have been revised.

Reviewer 2 Report

Comments and Suggestions for Authors

Author Response

Issues raised by Reviewer 2:

Comments and suggestions:

The research paper titled "Temporal variation of soil resistance to rill erosion in cropland of the dry-hot valley region, Southwest China" effectively encapsulates the focus of the research. It succinctly highlights the temporal aspect of soil erosion resistance in croplands of a specific geographical area, which is crucial for drawing attention to the dynamics of erosion processes in response to changing environmental conditions. However, some specific points could enhance the clarity and robustness of the paper.

Abstract:

The authors effectively communicate the significance of their findings in terms of enhancing our understanding of soil erosion dynamics in croplands, particularly in dry-hot climate regions like the dry-hot valley region of Southwest China.

Introduction:

The introduction provides a comprehensive overview of the background literature and sets the stage for the research by highlighting the significance of studying soil erosion in croplands, particularly in the context of global food production and land resource sustainability.

While this section provides a comprehensive overview of the factors influencing soil erosion resistance, there are instances where the information could be streamlined for clarity and conciseness. For example, some sections contain multiple citations in quick succession, which could be consolidated or integrated to improve the flow of the narrative.

Reply and revision: Thanks for your comments and suggestions. We have improved the introduction section according to your and other reviewer’s comments. The sentences in L50-57, L63-64, L73-76, L83-85, L92 and L101-102 have been revised and some reference citations have been removed to improve the flow of the narrative.

The introduction briefly mentions the research hypothesis regarding the interaction of seasonal dry-wet alternation, tillage disturbances, and crop growth in shaping soil erodibility dynamics. Providing a more explicit statement of the specific hypotheses being tested would enhance the clarity of the research objectives and facilitate the interpretation of the study findings.

Reply and revision: Thanks for your comments. We have improved the hypothesis and the novelty according to your and other reviewer’s comments (L109-110, L116-119).

Materials and Methods:

However, there are a few areas where clarification or expansion would strengthen the

While this section mentions the establishment of nine cropland plots with three treatments, it would be beneficial to briefly elaborate on the rationale behind these treatments and how they represent different management practices commonly observed in the study area.

Reply and revision: Thanks for your comments and suggestions. The three treatments in this study are not entirely based on local cropland management practices but also to show different mechanisms that determine soil erosion resistance. We have explained the purpose of the three treatments in Table 1. That are, the continuous fallow is to reflect the influence of the natural dry-wet cycle on soil detachment and resistance, fallow after tillage is to reflect the effect of tillage disturbances and the following consolidation effect on the soil resistance dynamics, tillage with corn it to reflect the effect of tillage disturbances and crop growth on soil resistance dynamics. To make sure the readers could understand the experimental design, we have added more contents in L165-166.

It would be beneficial to include additional details on soil characteristics such as texture, pH, and nutrient content, as these factors can influence soil erosion processes and resistance.

Reply and revision: Thanks for your comments and suggestions. We have added soil texture information according to your and other reviewer’s suggestion in L154-155.

Include the rationale behind selecting specific slope gradients and treatment combinations, particularly regarding their representativeness of real-world agricultural practices and their relevance to the study objectives.

Reply and revision: Thanks for your comments and suggestions. We have explained the design of the three different treatments in Table 1 and have added more contents in L165-166. Meanwhile, we also made revisions and added a reference citation [29] in L159 to support the selecting of slope gradients in this study.

Include details on the calibration of flow rates and the rationale behind selecting specific flow rate ranges.

Reply and revision: Thanks for your comments and suggestions. In L196-197 we have explained the flow rates were selected according to Cao et al. (2015), and these flow rates were within the range of the runoff generation capability recorded by local erosive storm studies. To help readers understand our experimental design, we have added more details about the calibration of flow rates in L198-199.

Results

Add specific numerical values or percentage changes in soil property indexes (e.g., bulk density, organic matter content, MWD) to quantify the magnitude of the observed differences between treatments and over time.

Reply and revision: Thanks for your comments and suggestions. We have added specific numerical values in section 3.1.

Please add correlation coefficients or regression analyses results to quantify the strength and direction of these relationships.

Reply and revision: Thanks for your suggestion. We have calculated the correlation coefficients between soil properties indexes including organic matter content, MWD, bulk density and shear strength. We found that only the positive correlation between bulk density and shear strength is significant at the 0.05 level. Therefore, we have added contents in L287-288: Furthermore, there is a positive correlation (r=0.758, p = 0.049) between the above two indexes.

Discussions

Here are some points to enhance the discussion.

Write mechanistic explanations for the observed changes in soil properties (e.g., the role of raindrop impact, soil particle binding, and vegetation) enhances the understanding of the results.

Reply and revision: Thanks for your comments and suggestions. We have improved the discussion section according to your and other reviewer’s comments. Contents in L388-389, L398-399, L406-407, L412-413, L421-426, L436-438, L454-456, L463-465 have been revised.

Conclusion

The conclusion needs to be rewritten. The conclusion should answer the research purpose of the manuscript and be a highly summarized sentence.

Reply and revision: Thanks for your comments and suggestions. We have rewritten this section and made sure the main conclusions can answer: (1) the temporal dynamics of soil erosion resistance; (2) the relationships between soil resistance and soil properties; and (3) the main mechanisms that determine the dynamics of soil resistance.

Reviewer 3 Report

Comments and Suggestions for Authors

GENERAL

I have doubts regarding this paper. On the one hand, the experiment seems to be well designed and results are interesting. On the other hand, I found serious issues, which must be amended before publication.

1) The information provided in 2. Materials and Methods section is insufficient:

a) the lack of information on particle size distribution of the soil (both before experiment and of the detached soil), or, at least, soil texture class is very important issue, because the soil texture is a key factor of soil erosion. The susceptibility to erosion of clayey and silty soils is completely different. The knowledge of this parameter is crucial for the interpretation and discussion of results of experiment. I hope the authors have soil samples to perform this analysis and complete this interesting paper!

b) I recommend to add information - code - of the climate type, e. g. according to Koppen-Geiger classification (please check https://koeppen-geiger.vu-wien.ac.at/ as example) or other climate classification, which is preferred by authors with particular reference. The information on the source of meteorological data (line 119) would be also welcome.

c) the information on soil reference group WRB - Lixisols - is quite contradictory with the later information on the „... laterite with high expansibility ...” (line 355, and line 475). The Lixisoils are shortly defined as „Soils with clay-enriched subsoil (with) low-activity clays, high base status” (please, check pdf available at https://obrl-soil.github.io/wrbsoil2022/index.html and page 22 of this pdf). The laterite may contain clay minerals of high activity and expansibility (smectite and illite), but the most common clay mineral present in laterite is caolinite (https://geologyscience.com/geology-branches/mining-geology/lateritic-deposits/), which is not very expansible. For this reason, the authors should check, if the soils in concern are really lixisols with laterite as parent materials.

2) The other important issue is that the experiment was conducted in one-season (2019) . As I mentioned, the same experimental design seems to be adecuate, so this issue is much less limiting, from the scientific point of view, then the deficient information regarding particle size distribution of the soil and eroded material.

There are also some mainly editorial issues, which might be amended. Some of them I mention below.

In conclusion, I recommend major revision of this paper before publication.

DETAILED

Line 48: „... soil texture and fragment content... - please clarify the term „fragment” - do authors mean „rock fragments” gravel and stones) or other soil particles?

Line 118 or it’s proximity: the diagram showing the distribution of average temperature and (natural) sum of precipitations woul be welcome and it would facilitate the interpretation of this paper, even if the experiment itself was related to the artificial water addition (line 173).

Line 128: a reference regarding WRB and it’s version (year or edition) is needed.

Line 163: the additional information on respective numeric code of the growth stage of corn at time of conducting the run-off experiments (August and October) would be welcome, it is also important for interpretation and discussion. The Zadok’s or BBCH scale is most commonly used (https://plantevaernonline.dlbr.dk/cp/Documents/bbcheng.pdf)

Line 205: I don’t undestant the term „flow depth”, please clarify.

Line 224: Please, check if the „... weighted diameter of water stable aggregates (MWD, mm) would not be more approppriate in this case.

Line 288: What is the meaning of „ANOVA = 0,029”? Is this a p-Value obtained as a result of analysis of variance (ANOVA) or something else?

Figure 3: Please, add information on the statistical significance on the difference showed at each diagram.

Table 4: This table is not very comprehensive - the designations of R, SIG. and N should be explained. In my opinion, the title of table should be changed to „The results of correlation analysis ...”. I suppose that correlation coefficients are presented in line R (most commonly the lower case „r” is used), and N is for number of observations (records), most commonly designed also by the lower case letter „n”. I can also only suppose, that SIG. stands for p-Value, but I may not be right. Please, clarify, and add information on the asterisc („*”) meaning (statisticaly significant?).

Line 401: The authors could check, what is the particle size distribution of loess soil, studied by Liu et al (2023), compare it with the same parameter of their soil and thus this part of discussion would be much more relevant.

Line 411: What fertilizer (name, nature - mineral or organic, it’s composition) and at which dosis and and date was it applied? This information could be added to Materials and Methods and they would be very useful in this part of discussion.

Line 440: And what is the soil texture class of studied soil?

Author Response

Issues raised by Reviewer 3:

GENERAL

I have doubts regarding this paper. On the one hand, the experiment seems to be well designed and results are interesting. On the other hand, I found serious issues, which must be amended before publication.

1) The information provided in 2. Materials and Methods section is insufficient:

  1. a) the lack of information on particle size distribution of the soil (both before experiment and of the detached soil), or, at least, soil texture class is very important issue, because the soil texture is a key factor of soil erosion. The susceptibility to erosion of clayey and silty soils is completely different. The knowledge of this parameter is crucial for the interpretation and discussion of results of experiment. I hope the authors have soil samples to perform this analysis and complete this interesting paper!

Reply and revision: Thank you very much for your comments and suggestions. We apologize for the lack of information and fully agree with your opinions about the importance of soil texture. During the experiments, the surface soils were sampled and properties including soil particle size were measured for all field plots. The specific soil texture data have been published in our previous paper (Zhao, Y.Y., Ruan, J.R., He, W., Wang, Y., Zhao, S.L. 2023. Study of soil infiltration capability dynamic in typical slope farmland of the Liangshan dry-hot valley region. Journal of Sichuan Normal University (Natural Science), 46(3): 398-405.). We found that there was no significant difference in soil particle size between the three treatments (please see the table below). Therefore, in this study we did not consider soil texture as an influencing factor in the variation of soil erosion resistance. However, soil texture information is still very important for understanding soil erosion resistance and should be provided. To avoid the similar content appearing in different papers, we have summarized the information in the table and have added soil texture information in L154-155 of the revised manuscript: The soil texture in the top 20 cm of the field plot is clay loam (USDA classification), with 34.33-36.73% clay, 21.60-24.67% silt and 41.00-42.00% sand.

Soil particle size of different treatments

Treatments

Clay (%)

Silt (%)

Sand (%)

Continuous fallow

34.33

24.67

41.00

Fallow after tillage

34.73

23.27

42.00

Tillage with corn

36.73

21.60

41.67

  1. b) I recommend to add information - code - of the climate type, e. g. according to Koppen-Geiger classification (please check https://koeppen-geiger.vu-wien.ac.at/ as example) or other climate classification, which is preferred by authors with particular reference. The information on the source of meteorological data (line 119) would be also welcome.

Reply and revision: Thank you very much for your suggestion. We have added more information about the climate type according to Koppen-Geiger classification in L128-129: The climate is warm temperate with winter dry and warm summer (Cwb according to the Koppen Geinger classification). Meanwhile, we also added a new reference citation [30] of the meteorological data source.

  1. c) the information on soil reference group WRB - Lixisols - is quite contradictory with the later information on the „... laterite with high expansibility ...” (line 355, and line 475). The Lixisoils are shortly defined as “Soils with clay-enriched subsoil (with) low-activity clays, high base status” (please, check pdf available at https://obrl-soil.github.io/wrbsoil2022/index.html and page 22 of this pdf). The laterite may contain clay minerals of high activity and expansibility (smectite and illite), but the most common clay mineral present in laterite is caolinite (https://geologyscience.com/geology-branches/mining-geology/lateritic-deposits/), which is not very expansible. For this reason, the authors should check, if the soils in concern are really lixisols with laterite as parent materials.

Reply and revision: Thank you very much for your comments and suggestions. The soil type in this study could be classified as red soil according to the Genetic Soil Classification of China (GSCC). However, it is difficult to precisely define the soil in the WRB system because the red soil in GSCC classification could refer to many different WRB soil types. In our pervious manuscript, we recognized the soil type as the Lixisols according to an approximate reference system provided by Zhou and Shen (2013) (Zhou, J.M., Shen R.F. 2013. Dictionary of Soil Science, Science Press, Beijing, Page778.). In order to define the soil type more precisely, we have overlaid the study site to the WRB 2015 World Soil Map (https://files.isric.org/public/WRB/WRB2014_soil_map.zip). We found that the study site is within the area of Alisols (please see the figure below). Therefore, we have revised the soil type as Alisols (L136-137 of the revised manuscript) which is characterized as the high-activity clays.

The figure is in the attached file.

Figure. The study site as overlaid with the WRB 2014 World Soil Map

AL: Alisols, AC: Acrisols, LV: Luvisols, AT: Anthrosols, FL: Fluvisols

Moreover, the term “dry laterite” in L355 of the previous manuscript might be a misuse of the traditional Chinese name (the dry red soil) for the local soil in the dry-hot valley conditions. We described the soil parent materials in the previous manuscript as the weathered granitic materials according to the Soil Species of Sichuan, which recorded the soil survey results during the 1980s based on the GSCC (Soil Survey Office of Sichuan Province. Soil Species of Sichuan; Sichuan Science and Technology Press: Chengdu, China,1994.). To further clarify the soil parent materials, we reviewed the Soil Series of China (Sichuan volume), which is based on the Chinese Soil Taxonomy (CST) and is the most recent and currently most accurate basic soil data in China (Yuan, D.G.; Zhang, G.L. Soil Series of China, Vol. Sichuan. Science Press, Beijing, China, 2021.). We have found a typical soil profile close to our study site and the soil parent material is the Quaternary ancient red soils. This type of soil parent material was reported by He et al. (2008) as highly expansible. We have revised the related contents in L136 and L388 and added a reference [31].

2) The other important issue is that the experiment was conducted in one-season (2019). As I mentioned, the same experimental design seems to be adecuate, so this issue is much less limiting, from the scientific point of view, then the deficient information regarding particle size distribution of the soil and eroded material.

Reply and revision: Thank you very much for your comments. The experimental design of this study covered the entire rainy season, which accounts for about 90% of the annual rainfall. The three-time stages were selected according to the corn growth process and coincided with the beginning, middle and end of the rainy season. It could therefore reflect the interaction effect of seasonal wet-dry cycle, tillage disturbance and crop growth. To give readers more information about our experimental design, we have revised contents in L184-186 to show the relationship between wet season and corn growth stage.

DETAILED

Line 48: ... soil texture and fragment content... - please clarify the term “fragment” - do authors mean “rock fragments” gravel and stones) or other soil particles?

Reply and revision: We apologize for the vague term. We have revised it more specifically as “rock fragment” in L49.

Line 118 or it’s proximity: the diagram showing the distribution of average temperature and (natural) sum of precipitations would be welcome and it would facilitate the interpretation of this paper, even if the experiment itself was related to the artificial water addition (line 173).

Reply and revision: Thanks for your comments and valuable suggestion. We have added a new figure (Figure 2) to show the monthly distribution of local temperature and precipitation. During the experiments of this study, the simulated runoff would pass through rills which areas are very small (0.1m wide and 4m long) compared to the plot area (5m wide and 9m long). In addition, the experiments were less than 10 min and most of the water would be sampled or drained off the plots. Therefore, artificial water addition may have a relatively small effect on the field plots compared to annual rainfall.

Line 128: a reference regarding WRB and it’s version (year or edition) is needed.

Reply and revision: Thanks for your suggestion. As we have determined the soil type according to the WRB 2015 World Map, we have added the reference with year information (WRB, 2015) accordingly.

Line 163: the additional information on respective numeric code of the growth stage of corn at time of conducting the run-off experiments (August and October) would be welcome, it is also important for interpretation and discussion. The Zadok’s or BBCH scale is most commonly used (https://plantevaernonline.dlbr.dk/cp/Documents/bbcheng.pdf)

Reply and revision: Thanks for your comments and valuable suggestion. We have added the information about corn growth stage in L184-188 respectively. Meanwhile, we also explained the effect of corn coverage in affecting soil properties in the discussion section in L454-456.

Line 205: I don’t understand the term “flow depth”, please clarify.

Reply and revision: We apologize for the vague term. We have revised it more specifically as “depth of the flow within the rill” in L230.

Line 224: Please, check if the „... weighted diameter of water stable aggregates (MWD, mm) would not be more approppriate in this case.

Reply and revision: Thanks for your comments. We apologize for the missing word. The term should be “mean weight diameter of water stable aggregates”. We have added a reference citation [40] about this term.

Line 288: What is the meaning of “ANOVA = 0,029”? Is this a p-Value obtained as a result of analysis of variance (ANOVA) or something else?

Reply and revision: We apologize for the mistake. It should be “ANOVA, p = 0.029”. We have made revisions in L314 and L318 respectively.

Figure 3: Please, add information on the statistical significance on the difference showed at each diagram.

Reply and revision: Thank you very much for your suggestion. We have revised this figure (Figure 4 in the revised manuscript) accordingly.

Table 4: This table is not very comprehensive - the designations of R, SIG. and N should be explained. In my opinion, the title of table should be changed to „The results of correlation analysis ...”. I suppose that correlation coefficients are presented in line R (most commonly the lower case „r” is used), and N is for number of observations (records), most commonly designed also by the lower case letter „n”. I can also only suppose, that SIG. stands for p-Value, but I may not be right. Please, clarify, and add information on the asterisc („*”) meaning (statisticaly significant?).

Reply and revision: Thank you very much for your detailed comments and suggestions. We have revised Table 4 accordingly.

Line 401: The authors could check, what is the particle size distribution of loess soil, studied by Liu et al (2023), compare it with the same parameter of their soil and thus this part of discussion would be much more relevant.

Reply and revision: Thank you very much for your comments. We have added soil particle size and texture information of the loess soil (L437-438). Meanwhile, the reference Liu et al (2023) has been changed into Liu et al (2024) according to the updated information. It should be noted that the results of Liu et al. (2024) are based on the hydraulic flume experiments, which makes the soil detachment process different from our field runoff simulation experiments. Therefore, we compared the decrease rate instead of directly comparing the values of rill erodibility. We have made revision in L436-437 to show the experiment method of Liu et al. (2024).

Line 411: What fertilizer (name, nature - mineral or organic, it’s composition) and at which dosis and date was it applied? This information could be added to Materials and Methods and they would be very useful in this part of discussion.

Reply and revision: Thank you very much for your comments and suggestions. We have added more information in L171-173: The plots were fertilized at the same time as corn sowing with 90 kg/hm2 of N, 90 kg/hm2 of P2O5, 90 kg/hm2 of K2O, and 500 kg/hm2 of decomposed manure, in accordance with to the local requirements for corn growth [33].

Line 440: And what is the soil texture class of studied soil?

Reply and revision: Thank you very much for your comments. We have added soil texture information and the studied soil texture is also clay loam.

The authors are grateful for the reviewer’s professional comment and suggestions. Especially for the detailed documents and websites, which provided us great convenience in improving our manuscript. Thank you very much!

Round 2

Reviewer 1 Report

Comments and Suggestions for Authors

All comments were fully considered or revised. The current version can be accepted for publication by Land.

Author Response

Comments from Reviewer 1:

All comments were fully considered or revised. The current version can be accepted for publication by Land.

Reply: Thank you very much for your comments. We have made further minor revisions according to other reviewer’s suggestions.

Reviewer 2 Report

Comments and Suggestions for Authors

Having thoroughly reviewed the manuscript, I recommend its acceptance as the author has diligently incorporated all the suggested changes, significantly improving the overall quality and clarity of the work.

Author Response

Comments from Reviewer 2:

Having thoroughly reviewed the manuscript, I recommend its acceptance as the author has diligently incorporated all the suggested changes, significantly improving the overall quality and clarity of the work.

Reply: Thank you very much for your comments. We have made further minor revisions according to other reviewer’s suggestions.

Reviewer 3 Report

Comments and Suggestions for Authors

GENERAL

The manuscript was improved considerably in this version, as the authors addressed major part of my observations.

However, some of my notes have not been considered yet in the version I received and downloaded. I refer particularly to decimal codes regarding growth stages of maize, which should be mentioned in lines 177 (germination) and line 181 (ripening).

Additionally, the authors should provide more information on the fertilizers used in the experiment - was it one mineral compound fertilizer containing N, P and K, or 2 or 3 (simple) fertilizers? In any case, the authors should provide information on the name(s) and composition (percent content of M, P2O5 and K2O in it or them) of fertilizer(s). I have serious doubts regarding the dose of decomposed manure - it amounted really 500kg/hm2? I understand manure as a substance of animal origin - feaces and urine - mixed with straw or other substance used as litter. Such manure contain rather small amounts of nutrients, most frequently about 0,3-1,0% of N and respective contents of other nutrients. In my opinion it would be very difficult and unjustified to apply such manure dose to soil. The form of mineral fertilizer may have specific effect on soil structure and crust formation. The dose of manure affects humus content and soil structure, and, consequently, soil erodibility.

I have some smaller notes, which may be easily addressed before publication.

I recommend minor revision of this paper.

DETAILED

Line 122: Please, add respective reference regarding Koppen-Geiger climate classification - paper or link.

Line 137: I understand that traditional hoeing is manual?

Line 165: The authors could provide information on the plant population in hectare (hm2), it would be more relevant for agronomists.

Lines 168-169: Is weed destruction in fallowed land a common agricultural practice in the area of study? According to my knowledge, the main reason for land fallowing is soil regenration, which require plant (even weed) cover! The fallowing of land without cover is sometimes practices, but most frequently in deser areas to preserve soil moisture before sowing, but the study area is not a desert.

Table 2: please, shift it to current line 282, as it belongs to Results section.

Author Response

Issues raised by Reviewer 3:

GENERAL

The manuscript was improved considerably in this version, as the authors addressed major part of my observations.

However, some of my notes have not been considered yet in the version I received and downloaded. I refer particularly to decimal codes regarding growth stages of maize, which should be mentioned in lines 177 (germination) and line 181 (ripening).

Reply and revision: Thank you very much for your comments. We apologize for misunderstanding your suggestions in the previous revision. We have added the decimal codes regarding corn growth stages in L179-180, L181-182 and L183.

Additionally, the authors should provide more information on the fertilizers used in the experiment - was it one mineral compound fertilizer containing N, P and K, or 2 or 3 (simple) fertilizers? In any case, the authors should provide information on the name(s) and composition (percent content of M, P2O5 and K2O in it or them) of fertilizer(s). I have serious doubts regarding the dose of decomposed manure - it amounted really 500kg/hm2? I understand manure as a substance of animal origin - feaces and urine - mixed with straw or other substance used as litter. Such manure contain rather small amounts of nutrients, most frequently about 0,3-1,0% of N and respective contents of other nutrients. In my opinion it would be very difficult and unjustified to apply such manure dose to soil. The form of mineral fertilizer may have specific effect on soil structure and crust formation. The dose of manure affects humus content and soil structure, and, consequently, soil erodibility.

Reply and revision: Thank you very much for your suggestion. In this study, a compound fertilizer provided by the Stanley Agriculture Group Co., Ltd. was used. We have added the name and NPK percent content of the mineral compound fertilizer, and have revised the original contents more clearly as “600 kg/hm2 of Stanley compound fertilizer (N:P2O5:K2O = 15:15:15)” (L166 of the revised manuscript). Meanwhile, we apologize for the writing mistake in the manure amount. It should be 5000 kg/hm2 rather than 500 kg/hm2 (L167 of the revised manuscript).

We agree with your opinions that the mineral fertilizer can affect soil structure and crust formation. In the case of this study, the fertilizer is essential for the corn to grow, which in turn can affect soil erosion resistance. The effect of mineral fertilizer itself on soil properties and erosion resistance is not the main objective of this study. Moreover, the relatively low bulk density and shear strength for the tillage-with-corn treatment indicate that soil crust formation caused by mineral fertilizer may not be significant when compared to the fallow-after-tillage treatment which received no fertilizer. We have discussed the different role of soil crust in affecting the erosion resistance of the corn-planted soils in L450-454. Meanwhile, we also discussed the effect of manure application in affecting soil properties and erosion resistance in L438-440 and L454-461, respectively.

DETAILED

Line 122: Please, add respective reference regarding Koppen-Geiger climate classification - paper or link.

Reply and revision: Thank you very much for your suggestion. We have added the link in L123-124.

Line 137: I understand that traditional hoeing is manual?

Reply and revision: Thank you very much for your comment. The traditional hoeing is manual and we have added the word “manual” in L161.

Line 165: The authors could provide information on the plant population in hectare (hm2), it would be more relevant for agronomists.

Reply and revision: Thank you very much for your suggestion. We have made revision and added the content “67000 plants/hm2” in L165.

Lines 168-169: Is weed destruction in fallowed land a common agricultural practice in the area of study? According to my knowledge, the main reason for land fallowing is soil regenration, which require plant (even weed) cover! The fallowing of land without cover is sometimes practices, but most frequently in desert areas to preserve soil moisture before sowing, but the study area is not a desert.

Reply and revision: Thank you very much for your comment. We agree with you that the traditional fallow practices are generally require plant cover. However, the experimental designs in this study are not entirely based on local agricultural management practices but also to show different mechanisms that determine soil erosion resistance. The three treatments were set to reflect a stepwise effect of natural dry-wet cycles, tillage disturbances, and crop growth on soil erosion resistance. We have explained the purpose of different treatments in L157-160 and in Table 1. That are, the continuous-fallow is to reflect the influence of the natural dry-wet alternation on soil detachment and resistance. In contrast, the fallow-after-tillage treatment reflects the effect of tillage disturbances and the following consolidation effect on the soil resistance dynamics during the dry‒wet alternation. The tillage-with-corn treatment is to reflect the effect of tillage disturbances overlaid with crop growth on soil resistance dynamics. In order to compare the results of continuous-fallow with the other two treatments, weeds should be removed from the fallow plots and ensure that soil resistance is mainly influenced by the natural dry-wet cycle. We have added content “and ensure that soil resistance is mainly affected by natural dry‒wet alternation” to explain this point more clearly in L170-171.

Table 2: please, shift it to current line 282, as it belongs to Results section.

Reply and revision: Thank you very much for your suggestion. We have made revisions accordingly (L284 of the revised manuscript).